# Research on online book user purchase behavior based on the event logic graph

**Bo Zhang**⊙*, **Shiling Peng**⊙ ⊙

College of Publishing, University of Shanghai for Science and Technology, Shanghai, China

⊙ These authors contributed equally to this work.
* zhangbo@usst.edu.cn

## Abstract

### Purpose/Significance

Consumer psychology and demand preferences embedded within user reviews constitute core intelligence resources for precise business operations. This study focuses on the online book consumption scenario, aiming to construct an analytical framework for online book user purchase behavior based on Event Logic Graphs (ELGs). This framework deeply analyzes the internal logical chains and pattern regularities within user behavior events. It seeks to expand the research boundaries of user behavior analysis and ELG applications theoretically, while simultaneously providing practical support for e-commerce platform intelligent operations and the publishing industry's precision marketing. Thus, it possesses both theoretical innovation value and application prospects.

### Methods/Process

Using Dangdang.com book reviews as the data source, the Top2Vec unsupervised topic clustering method was employed to extract user purchase behavior themes. Combining this with Gephi, an ELG was constructed where clustered themes served as nodes and semantic relationships between themes as edges. Visualization techniques were leveraged to deduce the logic behind user purchase behavior and uncover latent demand preferences.

### Results/Conclusions

Online book purchasing behavior exhibits a causal logical chain of "Motivation Triggering → Decision Implementation → Feedback Iteration": The motivation layer encompasses diverse demand orientations like cognitive enhancement and emotional connection; the decision layer is driven by multidimensional factors including product aesthetics, social trust, and price perception; the feedback layer forms a closed-loop mechanism involving quality supervision and emotional continuity. Based

**Data availability statement:** All relevant data are within the paper and its Supporting information files.

**Funding:** This work was supported by the National Social Science Fund of China (Grant No. 23BXW101) "Research on Publishing Big Data Mining and Utilization Based on Knowledge Graph".

**Competing interests:** The authors have declared that no competing interests exist.

on the behavioral characteristics revealed by the ELG, online book retailers need to anchor demand scenarios, building a precision operation system across four dimensions—content ecosystem, product form, marketing reach, and service quality control—to synergistically achieve growth in both user value and commercial value.

## Innovation/Limitation

The innovation lies in integrating Top2Vec theme clustering with ELG visualization technology, establishing a "semantic aggregation + logical deduction" research paradigm for consumer behavior. Limitations include the potential for small sample themes to weaken the explanatory power for group heterogeneity, and the research scope being currently confined to the "purchase behavior" stage without extending to the entire reading lifecycle. Future research should deepen conclusions by expanding data dimensions and scenario boundaries.

---

## 1. Introduction

The rapid development of internet technology and the increasing maturity of e-commerce platforms have established online book retailing as the dominant global channel for book consumption. Compared to traditional brick-and-mortar bookstores, online platforms attract massive users through rich assortments, convenient price comparisons, personalized recommendations, and efficient logistics. According to the "2024 Annual Report on China's Book Retail Market" by Beijing OpenBook, the total retail market scale reached ¥112.9 billion RMB in 2024, with platform e-commerce accounting for the largest share at 40.92% [1].

However, online book purchasing is not a simple "click-buy" instantaneous decision, but rather a dynamic, complex, multi-stage process. Within this process, users may undergo a series of events: need generation, search/browsing, information comparison, adding to cart/wishlist, order placement/payment, and subsequent evaluation/recommendation. These events are not isolated; they exhibit intricate logical connections and causal driving relationships [2]. For instance, a negative review might cause a user to abandon a purchase; a recommendation for a popular book might trigger purchases of related titles; browsing books on a specific topic might lead algorithmic recommendations to guide users towards deeper or broader knowledge domains. Traditional user behavior analysis methods, based on user profiles, collaborative filtering, or simple sequence models, struggle to effectively capture and parse these deep-seated, structured dynamic event sequences and their inherent causal logic. Concurrently, Event Logic Graphs (ELGs), an emerging technology in artificial intelligence and knowledge graphs, demonstrate a powerful capability for understanding and reasoning about complex event chains. Focusing on "events" as core nodes, ELGs link events through causal semantic relationships, constructing a structured knowledge network that reflects event evolution logic, causal chains, and contextual patterns. They excel at mining the underlying pattern logic hidden within vast amounts of behavioral sequence data, offering a novel approach and tool for

solving the problems of deep association and causal reasoning in online user behavior analysis [3].Compared to traditional sentiment analysis or topic modeling, the core value of event logic graphs lies in their ability to dynamically map causal relationships and temporal evolution between events, deconstruct heterogeneity in behavioral decision-making, and capture long-term feedback cycles. This capability enables event logic graphs to not only identify root causes of problems and strategic leverage points for intervention but also to reveal complex conditional dependency rules. Consequently, they provide actionable causal insights for designing precise optimization strategies, thereby advancing user behavior analysis from descriptive models toward predictive and intervention-oriented frameworks.

Therefore, this study aims to explore the establishment of an analytical framework for online book user purchase behavior based on ELGs. It seeks to deeply parse the internal logical chains and pattern regularities within user behavior events. This endeavor not only theoretically expands the research scope of user behavior analysis and ELG applications but also provides strong practical support for the intelligent operations of e-commerce platforms, user experience enhancement, and precision marketing within the book industry. Consequently, it holds significant theoretical innovation value and promising application prospects.

## 2. Related research

### 2.1. Research on online book purchasing behavior

In the study of online book purchasing behavior, scholars primarily employ methods such as Structural Equation Modeling (SEM) [4], Fuzzy-Set Qualitative Comparative Analysis (fsQCA) [5], and machine learning prediction models [2] to explore user decision-making mechanisms, characteristics of divergent consumer groups, and the effectiveness of marketing strategies. Research contexts are often centered on emerging channels like live-streaming marketing, algorithmic recommendations, and social platforms. In live-streaming marketing, Cheng et al., based on flow theory, validated through a dual-mediation model that streamer professionalism drives purchase intention via flow experience and brand attitude [6]; Pan et al., combining Convolutional Neural Networks (CNN) to quantify short video presentation styles, found that product-feature-focused strategies significantly boost conversion rates, while character-centered narratives have a negative impact [7]. Regarding user groups, Tang, grounded in the Theory of Planned Behavior, empirically demonstrated that the book consumption willingness of "small-town youth" is significantly driven by subjective norms and perceived behavioral control [8]; Su revealed that parental anxiety and educational expectations mediate the relationship between policy risk (e.g., "Double Reduction" policy) and supplementary textbook purchasing behavior [9]. Existing research also focuses on technology-enabled innovation in consumption scenarios. Zhang et al. used heterogeneous network representation learning to integrate multi-dimensional features like authors and keywords, improving book recommendation accuracy by 19.52% and significantly optimizing diversity [10]; Wang et al. verified that Augmented Reality (AR) technology reduces information inconsistency through vividness and interactivity, particularly boosting consumption of affordable books [11].

However, current research exhibits two main limitations. Firstly, methodological staticity often relies on cross-sectional surveys, like Yang et al.'s fsQCA study based on the AISAS model [12], lacking dynamic tracking and real-time evolution analysis of user behavior. Secondly, there is insufficient theoretical integration, such as fragmented factor analysis failing to construct a comprehensive "Need → Decision → Evaluation" behavioral logic chain.

### 2.2. Research on event logic graph applications

Event Logic Graphs (ELGs) are knowledge bases describing the evolutionary patterns and logical relationships between events [13,14]. Their core value lies in mining logical relationships such as succession and causality within unstructured event data, enabling the structured representation and predictive reasoning of dynamic event knowledge. Current domestic ELG application research shows characteristics of domain concentration and technical diversification. Domain-wise, it primarily focuses on public safety [15], online public opinion [16,17], and policy analysis [18]. For example, Zeng et al. [19]

integrated sentiment factors and event impact to construct an ELG for epidemic public opinion, revealing event propagation paths; Zhang et al. [20] used an ELG to analyze the multi-level transmission chain of US semiconductor policies towards China, assessing industrial chain risk evolution trends. Technically, a fusion approach combining rule templates with deep learning is common. Huang et al. [21] used ChatGPT prompt templates for zero-shot event extraction; Xiao et al. [22] employed the RoBERTa model to enhance causal event pair identification efficiency and introduced event generalization techniques, like BERTopic [23] theme clustering and word vector similarity calculation [24], to compress redundant information and build abstract ELGs. However, existing research suffers from limitations like rigid application scenarios, primarily focusing on fields like public opinion governance and emergency management, while neglecting the mining of event logic in the domain of user behavior information analysis.

User online purchasing behavior is essentially an event-driven dynamic evolution process [25]. Various stages are influenced by multiple event logics, such as policy stimuli (e.g., "Double Reduction" triggering anxiety about teaching supplementary materials) and technological interaction [26] (e.g., AR reducing information asymmetry) [27]. ELGs, through event node abstraction and relationship link deduction, are well-suited to deconstruct the internal evolutionary mechanisms of such behaviors. For instance, revealing the causal chain "live interaction → flow experience → impulse purchase" [28], or identifying the succession path "subjective norms drive consumption among small-town youth groups" [29]. In summary, this paper attempts to leverage ELG methodology to investigate online book purchasing user behavior. Firstly, product-related reviews from virtual communities are collected as the core data source. Secondly, rule-based template matching combined with dependency parsing NLP techniques is used to identify and extract events and their causal relationships. Finally, the Top2Vec event clustering model semantically aggregates events to generate high-level abstract event nodes, and the Gephi tool visualizes the causal ELG of user purchase behavior, enabling in-depth mining of online book consumer purchasing characteristics and evolutionary mechanisms.

## 3. Research methods and design

### 3.1. Selection of research object

Dangdang.com was selected as the research object based on three key considerations. Firstly, as China's earliest and leading vertical e-commerce platform specializing in books and audiovisual products, Dangdang has accumulated a vast review repository covering all categories, spanning extended time periods, and deeply reflecting users' genuine reading experiences and purchase decision processes. Secondly, its review content is detailed and rich; users frequently share purchase motivations, comparative considerations, usage effects, and emotional attitudes in depth, providing a high-quality data source with strong industry representativeness and information density for directly mining core drivers of purchasing behavior. Finally, Dangdang has long held a core position in China's online book retail market, with its market share consistently ranking among the top three, making it a typical representative of vertical book e-commerce. Therefore, this study selected Dangdang.com book reviews as the research data source, choosing 119 best-selling books from its 10 major vertical categories.

### 3.2. Selection of research object

Dangdang.com was selected as the research object based on three key considerations. Firstly, as China's earliest and leading vertical e-commerce platform specializing in books and audiovisual products, Dangdang has accumulated a vast review repository covering all categories, spanning extended time periods, and deeply reflecting users' genuine reading experiences and purchase decision processes. Secondly, its review content is detailed and rich; users frequently share purchase motivations, comparative considerations, usage effects, and emotional attitudes in depth, providing a high-quality data source with strong industry representativeness and information density for directly mining core drivers of purchasing behavior. Finally, Dangdang has long held a core position in China's online book retail market, with its market

share consistently ranking among the top three, making it a typical representative of vertical book e-commerce. Therefore, this study selected Dangdang.com book reviews as the research data source, choosing 119 best-selling books from its 10 major vertical categories.

Data were extracted using the web scraping tool "Octopus Collector" (Octopus Data Technology Co., Ltd.), configured to comply with Dangdang.com's robots.txt directives and to respect reasonable request rate limits, thereby minimizing server load. The collection and use of this data strictly adhered to Dangdang.com's Terms of Service, which explicitly permit the non-commercial, academic use of publicly displayed content, provided that no personal or sensitive information is harvested. Consistent with this policy, our collection process accessed only publicly visible review text; no personally identifiable information (such as usernames, purchase records, or contact details) was retrieved, stored, or utilized. All data were anonymized and aggregated at the comment level.

In accordance with Dangdang.com's privacy policy, the original raw review texts cannot be made publicly available. However, the processed and structured dataset—including 1,313 extracted causal event pairs—can be provided by the corresponding author upon reasonable request for research verification purposes.

### 3.3. Research framework

The online review mining process based on ELG primarily consists of five steps: Data Collection and Preprocessing, Causality Recognition and Event Extraction, Clustering Generalization, Graph Construction, and Result Analysis (Fig 1).

**3.3.1. Comment collection and preprocessing.** Dangdang.com, characterized by high product sales volume, extensive reviews, and deep specialization in the book market, served as the data source. The Octopus Collector tool was used to scrape review pages, capturing book titles, review text, and timestamps. The Jieba segmentation package and LTP NLP toolkit were employed for data preprocessing. Initially, invalid texts (duplicates, very short comments, containing only punctuation/emojis) were removed. Sentences were then segmented into clauses and stop words were filtered out. Finally, an effective review text corpus was constructed.

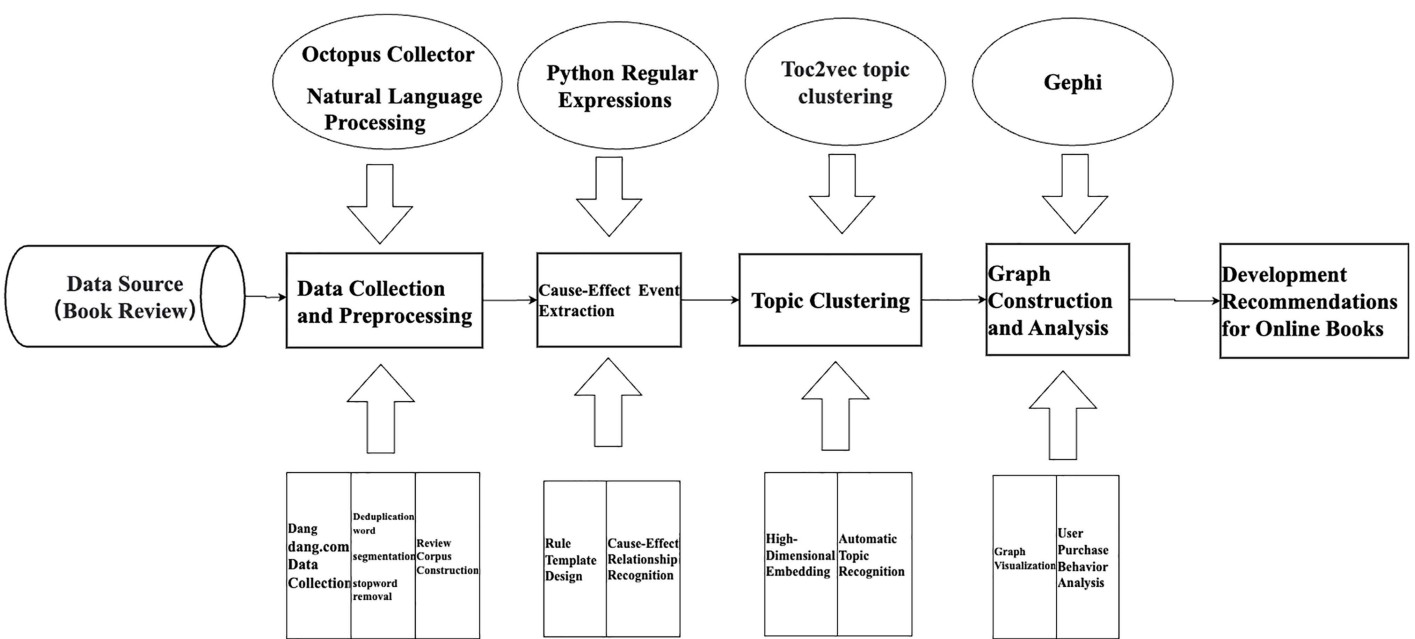

**Fig 1. ELG-based online review mining process.**

### 3.3.2. Causal event extraction.

Rule and template-based methods were adopted to identify commonly occurring causal relationships between events. Building upon the causal trigger word and template-based framework for causal event pair extraction by S. Zhao and Sorgente [30], and addressing systematic syntactic and expressive differences between Chinese and English [31], this study integrated the classification system for Chinese causal complex sentences proposed by Chu et al. [32] (Center-Conjoining, End-Dependent, Bi-Ended) to precisely locate cause and effect semantic segments within sentences. Accordingly, ten categories of Chinese causal rule-matching templates were designed (details in Table 1). Given the unstructured nature of Dangdang reviews often leading to sentence component omissions, where traditional Subject-Verb-Object (SVO) recognition performs poorly, the method by Shan et al. [33] was referenced. Events were represented lightweightly using combinations of core noun phrases and verb phrases. Regular expressions were designed for each template type to enable automated causality identification. For example, the template 【{Cause},<Conj...>{Effect}】 matches conjunctions like 【所以】 (suǒyǐ, "so"), corresponding to the Center-Conjoining type. Using this template, a causal event pair like 【theory combined with practice】 【so】 【good book】 could be matched. Here, "theory combined with practice" is identified as {Cause}, the cause event, and "good book" as {Effect}, the result event.

### 3.3.3. Top2Vec theme clustering.

Given the large scale and significant distribution sparsity of the causal event pairs obtained during extraction, and considering that ELGs primarily consist of nodes and directed edges where heterogeneous event pairs exhibit semantic co-occurrence at theme level, theme clustering on the extracted event set was necessary. Merging semantically homologous events into theme clusters not only reduces data dimensionality complexity to optimize graph structure clarity but also achieves semantic abstraction through event generalization at the cluster level. For complex causal scenarios like "one cause, multiple effects" or "one effect, multiple causes", this clustering process supports feature aggregation of similar event samples, providing a structured analytical foundation for causal mechanism induction and event evolution trend prediction.

In the ELG construction context focused on structured expression of event causal logic, introducing Top2Vec for theme clustering leverages its advantages in deep semantic mining and automatic theme discovery. By mapping event texts into a unified semantic vector space, Top2Vec effectively captures latent semantic associations between documents and automates the identification of hidden thematic structures within large-scale unlabeled data, thereby providing robust support for event classification, causal reasoning, and knowledge organization. Furthermore, this approach enhances information extraction efficiency and graph construction quality, making the resulting ELG more interpretable and practically valuable. Therefore, this study utilized the Top2Vec model for cluster analysis.

**Table 1. Examples of causal syntactic rules.**

| No. | Syntactic Pattern | Rule |
|---|---|---|
| 1 | Matching Type Causality (Explicit) | (?:because\|since\|now that)(.?)(?:for this reason\|so\|thus\|so that\|hence)(.?)[,　;!?] |
| 2 | Mid-positioned Causality (Explicit) | (.?)(?:thus\|so\|thus\|cause\|make\|thereby)(.?)[,　;!?] |
| 3 | Front-positioned Causality (Explicit) | (?:since\|based on\|according to\|as long as\|now that\|because\|if)(.*?),　;!?[,　;!?] |
| 4 | Verb-driven Causality (Implicit) | (.?)(?:lead to\|bring about\|produce\|contribute to\|cause\|arouse\|induce\|prompt\|trigger)(.?)[,　;!?] |
| 5 | Matching Type Retrospective Causality(Explicit) | (.?)(?:the reason why)(?:is because\|is due to\|stems from)(.?)[,　;!?] |
| 6 | Mid-positioned Retrospective Causality (Implicit) | (.?)(?:originate from\|out of\|depend on\|come from)(.?)[　;!?] |
| 7 | Preposition-guided Causality (Implicit) | (.?)(?:derive from\|come from\|be rooted in\|be based on)(.?)[　;!?] |
| 8 | Passive-triggered Causality (Implicit) | (.?)(?:be triggered\|be caused\|make\|cause)(.?)[　;!?] |
| 9 | Front-positioned Vague Causality (Implicit) | (?:through\|by\|rely on\|according to)(.*?),　;!? [　;!?] |
| 10 | Mid-positioned Vague Causality (Implicit) | (.?)(?:lest\|in order to avoid\|so that\|in order to\|for this purpose)(.?)[　;!?] |

**3.3.4. Graph interpretation.** Combining the clustered theme demands with cause and result events, cause events were set as source nodes and result events as target nodes, with causal relationships between them as directed edges. The Gephi tool was then used to generate a visualized directed graph. Finally, based on this graph, the purchasing behavior of online book buyers was analyzed.

## 4. Experiment

### 4.1. Comment collection and knowledge extraction

User review texts from the Dangdang.com platform were collected as the data source, sampling proportionally from best-selling books across major categories, resulting in a total of 113,499 data points. During preprocessing, duplicate reviews and very short reviews were first removed. The Jieba segmentation package and LTP toolkit were then used for word segmentation and sentence splitting of the review texts. Finally, a stop word list was referenced and augmented (e.g., adding terms like wākākā, "hiahiahia") to filter out stop words from the segmentation results. After this processing, 45,017 valid data entries remained. Using the rule templates defined in the previous section, each review sentence was processed for causality pairing and event extraction, yielding 1,313 causal event pairs. Some examples are shown in Table 2.

### 4.2. Top2Vec theme clustering

All processed causal event pairs were compiled into a single document for thematic clustering. To effectively extract semantic patterns from large-scale causal events, this study employs Top2Vec to perform unsupervised topic clustering separately on cause phrases and effect phrases. Specifically, each human-verified causal event contains one or more cause phrases and effect phrases. Rather than concatenating them into full sentences, these components were processed independently: each cause phrase was formatted as "Cause: [content]" and each effect phrase as "Effect: [content]", with each formatted phrase treated as a distinct document input for the Top2Vec model.

A sensitivity analysis was conducted on Top2Vec's key parameter min_topic_size. When this parameter ranged between 30–40, the number of topics (24–28) and average topic size (86–104) remained relatively stable, indicating structural robustness in the clustering results. The value min_topic_size = 38 was ultimately selected as it optimally balances semantic granularity with avoidance of noisy micro-clusters (Fig 2).

Based on the thematic clustering results, the identified topics primarily focus on the act of purchasing books, tangible attributes of books, targeted purchases by specific demographic groups, promotional incentives, reading experiences with evaluative feedback, and edition selection preferences, among other aspects, as presented in Table 3.

### 4.3. Graph construction

The Gephi tool was applied to construct a visual graph from the clustered themes and causal event pairs. After forming the causal ELG, the Fruchterman Reingold layout in Gephi was selected. Main cluster rendering using different colors

**Table 2. Examples of causal event extraction.**

| No. | Cause Event | Effect Event |
|---|---|---|
| 1 | Book Title | Purchased This Book |
| 2 | Stimulated Everyone's Interest in History | This Book Is of Great Significance |
| 3 | Learned Coffee Knowledge | Purchased Books |
| 4 | Violent Delivery | Damaged Book Corners |
| 5 | School Requirements | Purchased Books |
| 6 | Due to Lin Yutang | Purchased Books |
| 7 | Read This Book | Developed a Great Interest in Coffee |

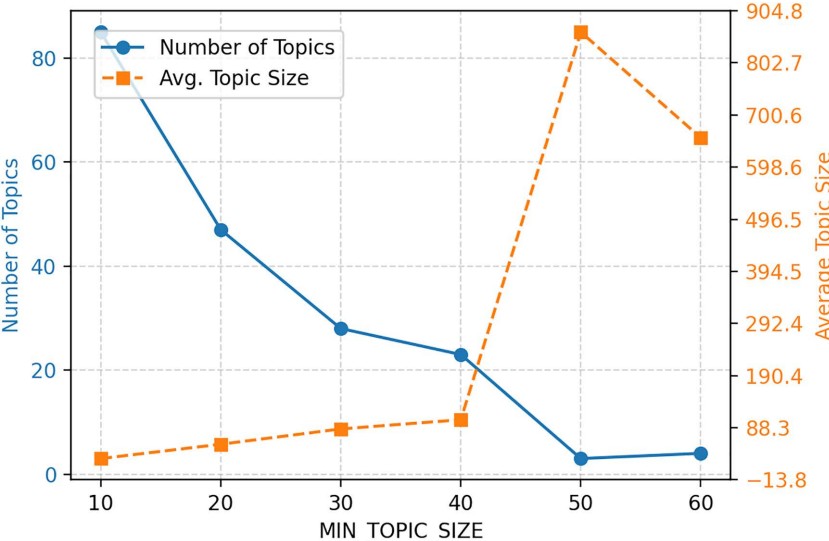

**Fig 2. Top2Vec parameter sensitivity analysis.**

**Table 3. Top 16 theme characteristic words.**

| Topic ID | Topic Name | Keywords |
|---|---|---|
| 1 | Fruit of Reflection | provoke deep thinking, stimulate thinking, benefit a lot, resonate, attract widespread attention, think of |
| 2 | Reading Books | read books, read this book, read the book, read this book, reading, must-read books |
| 3 | Elegant Book Design | well-bound book, read this book, buy books now, books, purchase this book, read this book, good book |
| 4 | Book Purchasing Guide | buy this book, buy books, purchase books, buy books now, purchase this book, buy paper books, read this book, buy many books |
| 5 | Repeat Purchase Preference | buy now, buy another copy, bought before, buy the third volume, discount, worth buying, buy paper version now, purchase books |
| 6 | Negative Review Experience | negative review, result, evaluation, write reviews, poor experience, comment, write comments, positive review, earn points, have an impact |
| 7 | history as a mirror | learn from history, this history book, take history as a mirror, era, attract widespread attention, read before, reason, provoke deep thinking, memory, author's unique ideas |
| 8 | Literary Discussion | learn about Su Dongpo, admire Liu Cixin's imagination, write reviews, learn from history, introduce, like Su Dongpo, attract widespread attention, see the big picture from the small |
| 9 | Discount Purchase Enjoyment | discount, cash redemption with points, reason, worth buying, benefit a lot, earn points, evaluation, buy now, result, negative review |
| 10 | Teacher Recommendation Benefit | teacher recommendation, resonate, benefit a lot, write reviews, colleague recommendation, provoke deep thinking, evaluation, stimulate thinking, rigorous content, introduce |
| 11 | Low Quality Causes negative comment | discount, inherent deficiencies in works, reason, full of, negative review, pirated books, works |
| 12 | Nostalgic Reasons | read before, read previously, reason, bought before, seen, attract widespread attention, stimulate thinking, think of |
| 13 | Love the Author | interest in the author, author's unique ideas, author, write reviews, this book, books, recommend this book, read this book, read the book |
| 14 | Must Read Children Book | teacher recommendation, read to children, must-read books, children finish reading, read books, read, read this book, books, reading, show to children |
| 15 | Children Fun Paradise | show to children, children finish reading, children read, children, read to children, teacher recommendation, stimulate thinking, write reviews, provoke deep thinking, see the big picture from the small |
| 16 | Digital Reading Memory | read electronic version before, read electronic version previously, seen electronic version before, read electronic version, read e-books, pirated books, bought pirated books before, buy paper version now, read before |

distinguished the themes. Each node represents an event. The edge connecting two nodes is a directed causal edge (e.g., Node1→Node2, where Node1 is the cause event and Node2 is the result event). As shown in Fig 3, Dangdang's online book purchasing behavior is constituted by causal events clustered under 16 closely related themes. Analysis of different nodes and causal logic within the graph reveals the needs, purchase behavior generation mechanisms, and evolutionary paths of online book purchasers.

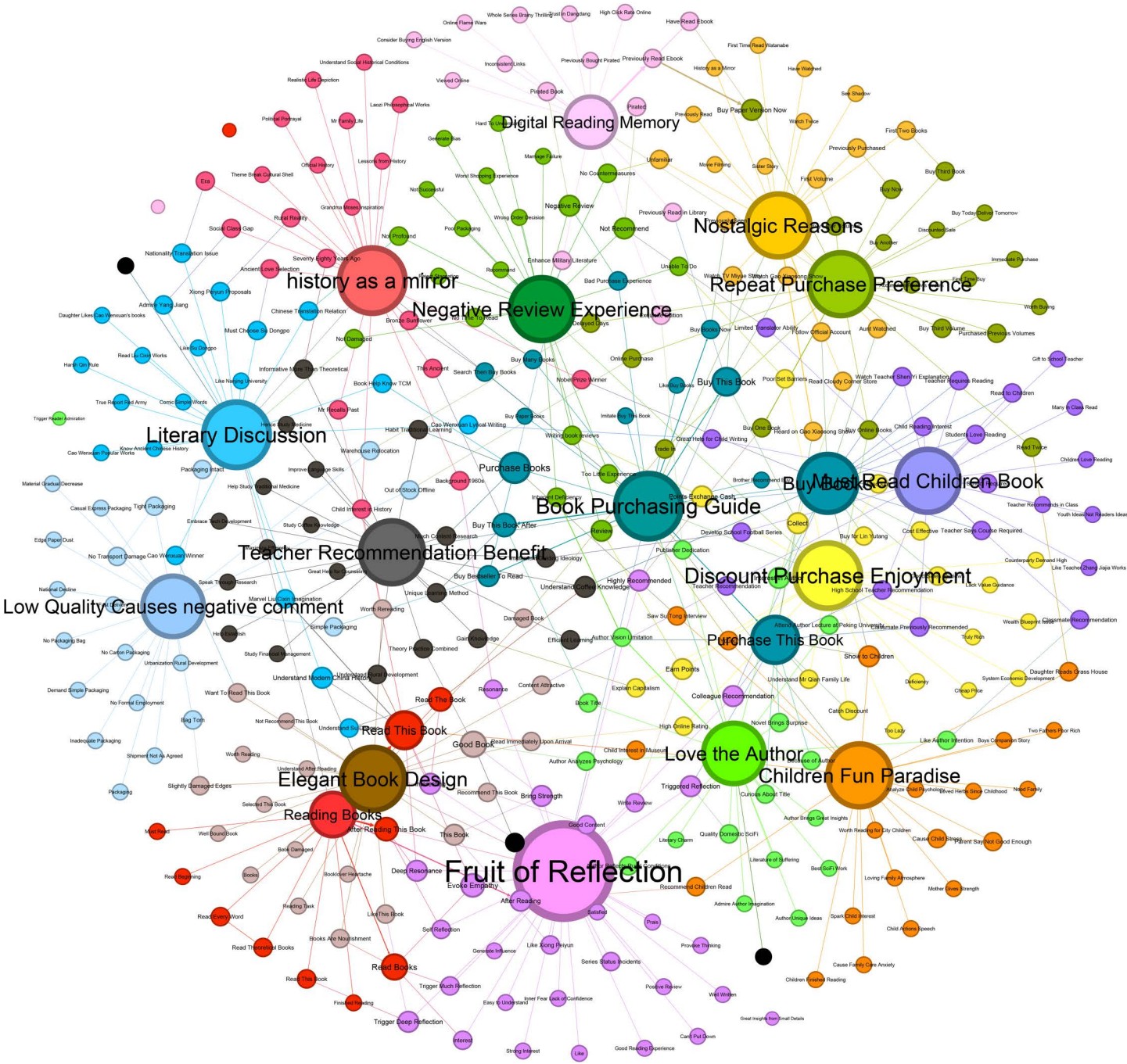

**Fig 3. Causal event logic graph of online book purchasing behavior.**

### 4.4. Graph interpretation

**4.4.1. Analysis of core nodes.** Observations of the graph reveal "Intellectual Resonance" and "Purchase Guidance" as core hubs both visually and structurally. Combining node layout, connection strength, and semantic labels, the core nodes and their associative logic are analyzed as follows."Purchase Guidance" appears as a large node at the center of the graph. It exhibits strong hub properties within a local sub-network through high-density connections to sub-node clusters like "bought this book," "purchase books," "bought this book because..." (Fig 4). Analyzing semantics and connection logic yields two key conclusions:

Decision-Driven Multi-Path: Sub-nodes form three behavioral branches around "Purchase Guidance":Social Trust: Nodes like "bought because big brother recommended," "imitated purchase" demonstrate the influence of friend/family recommendations and group trends ("circle following") on decisions. Social circle "trust endorsement" reduces user selection risk, becoming a non-commercial decision trigger.Such behaviors validate that "trust-based social proof" embedded in social relationships significantly mitigates users' book selection risks [34]. Crucially, while algorithm-driven recommendations dominate consumption contexts, this mechanism's non-transactional nature fosters stronger affective bonds with users.

Market Signal: Nodes like "came to read because it's a bestseller" reflect the guiding role of commercial labels like bestseller lists and platform recommendations. This indicates users rely on external market signals to "simplify" book selection, reducing information search costs.This contrasts with the "spontaneous impulse purchases" characteristic identified by Wang Yanyan et al. [35]. in live-streaming commerce research. Book consumption's market-signal dependency tends to induce content homogenization, necessitating vigilance against algorithmic suppression of content diversity.

Behavioral Characteristic: Nodes like "buy many books," "buy physical books," "searched and then bought books" dissect variations in purchasing behavior forms, reflecting personalized habits like "bulk buying," "preference for physical format," and "active searching."This phenomenon highlights how preferences for physical books epitomize the affective and archival value of print media, in contrast to the prevailing convenience-driven paradigm of digital consumption [36]. Such predilections align with the "scarcity of experiential consumption" theory in cultural economics [37].

Cross-Node Value Coupling: "Purchase Guidance" connects upwards to nodes like "Intellectual Resonance," "Teacher Recommendation & Shared Benefit," "Must-Read Children's Books," "Children's Fun Land," "Nostalgic Reasons," "Discount & Premium Enjoyment," "Author Influence," and "Mirror of History." It points directly downwards to behavior nodes like "buy books," "purchase this book," forming a complete chain: Preceding Influences (Content/Price) → Decision Filtering (Purchase Guidance) → Action Implementation (Buy Books). This structure proves that purchase decisions involve a collaborative filtering process of multi-dimensional signals (content value, price stimulus, social trust, scenario demand), rather than being a linearly driven single-factor behavior.However, traditional planned behavior theory underrepresents the complexity of social-market dynamics in the digital era [38], particularly overlooking cost-conscious segments' heightened sensitivity to perceived price-piracy associations.

"Fruit of Reflection" appears as a core node, acting as an aggregation hub for post-reading value output and behavioral extension. With high-density connections to over 30 semantic sub-nodes like "can't put it down," "provokes thought," "creates resonance," and to core nodes like "Book Reading," "Purchase Guidance," and "Aesthetic Binding Preference" (Fig 5), it constructs a network structure centered on "Intellectual Resonance," radiating to reading experience, purchase decisions, and cultural ritualistic feelings. This layout visually demonstrates that "Intellectual Resonance" is the energy transfer station where reading behavior extends into subsequent value chains. Emotional touches, cognitive reconstruction, and social interactions triggered by reading are all translated from experiential perception into behavioral drivers through "Intellectual Resonance." Sub-nodes around it can be grouped into four dimensions:Emotional Resonance: Nodes like "can't put it down," "like," "deeply resonate" anchor the value of books as emotional carriers, revealing user empathy with stories and emotional tones as a core source of reading stickiness.Distinct from research on aesthetic-driven visual stimuli dominant in cover design studies [39], our event logic graph demonstrates that content-triggered empathy

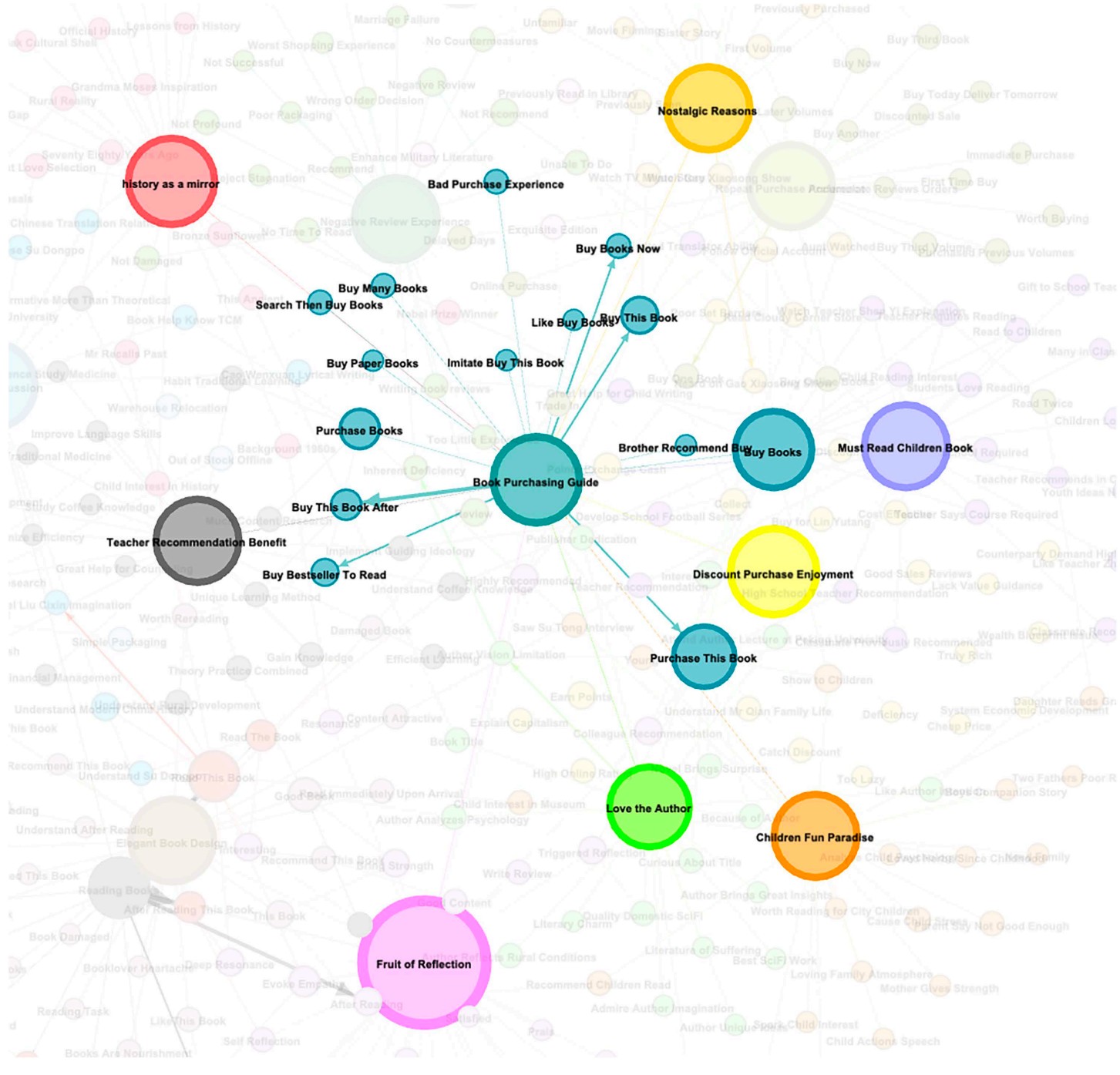

**Fig 4. Nodes Associated with "Purchase Guidance".**

constitutes the primary driver of reading engagement persistence. This finding resonates with the core affective priming mechanisms posited by cultural consumption theory.

Cognitive Reconstruction: Nodes like "provokes thought," "induces deep thought," "see the big from the small" reveal the impact of reading on restructuring users' cognitive frameworks, proving deep reading is essentially a knowledge

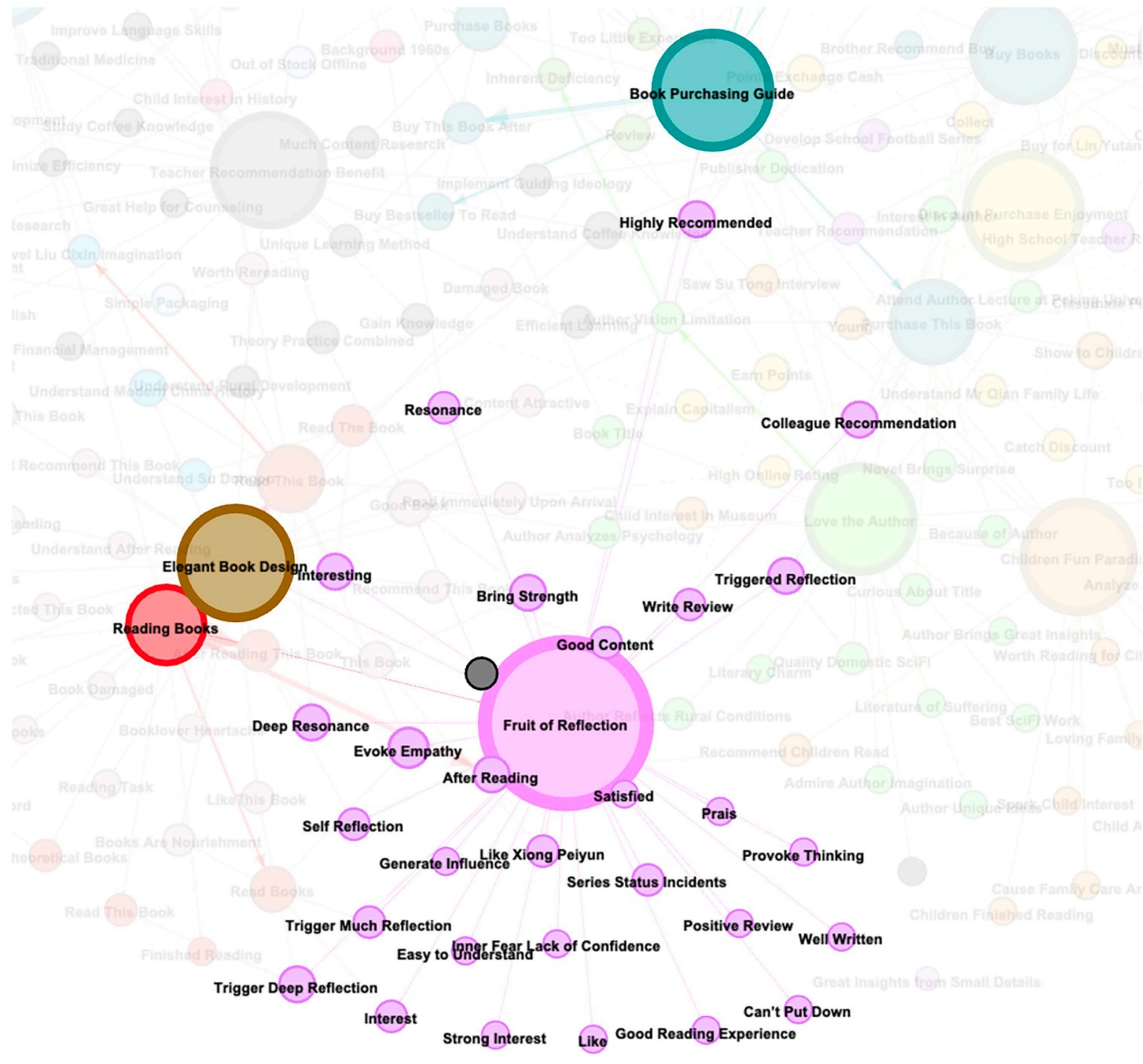

**Fig 5. Nodes associated with "intellectual resonance".**

reproduction process. Content Evaluation: Nodes like "good content," "easy to understand," "grew fond of Xiong Peiyun" both retroactively validate pre-purchase decisions (e.g., high-quality content drives deep resonance) and provide emotional anchors for repurchase loyalty (e.g., preference for authors/series). Social Propagation: The connection between "colleague recommendation" and "Purchase Guidance" embodies users' desire to recommend based on reading gains, turning books into social currency, driving the closed loop "Read → Resonate → Socially Share → Re-purchase.

Furthermore, "Intellectual Resonance" weaves multi-dimensional behavioral loops through its connections. Upwards linkage with "Book Reading" shows edges like "easy to understand," "triggered self-reflection" reveal reading quality directly influences resonance depth. Connection rightwards to "Purchase Guidance" forms an implicit path; for instance, recommendation desires sparked by resonance can transform into motives for purchase decisions within social circles. Downwards linkage to "Aesthetic Binding Preference" reflects the collection logic in nodes like "good content"; users might be driven by profound resonance to seek cultural ritualistic feelings (e.g., demand for beautiful bindings or collector's editions increases after deep engagement with quality content).

**4.4.2. Holistic analysis.** Analyzing the causal relationships among theme events in Dangdang's online purchasing behavior allows parsing users' internal logic and demand characteristics across four dimensions: reading motivation, purchase decision, feedback behavior, and group heterogeneity.

**4.4.2.1 Multi-Dimensional reading motivation.** Motivations manifest as three core orientations: intellectual enlightenment, historical insight, and emotional connection. Firstly, the need for intellectual stimulation is shown through the pursuit of mental resonance and cognitive expansion after reading, reflecting users' tendency to view books as vehicles for enlightenment and value reconstruction, often aligned with preferences for philosophical or social science texts. Secondly, historical insight motivation focuses on the cognitive tradition of "learning from history," where users explore practical reference points through historical texts, demonstrating the empowering value of history for real-world decisions. Thirdly, emotional connection motivation manifests as affective projection onto specific creators or cultural symbols, such as purchasing driven by identification with Su Dongpo's literary style or Liu Cixin's sci-fi narrative, essentially an extension of "fandom economy" into book consumption, reflecting the driving role of creator charisma on choices.

**4.4.2.2 Multi-faceted purchase decision mechanism.** Purchase decisions are jointly driven by three intertwined factors: product attributes, social trust, and price perception. Regarding product attributes, physical aspects like binding design and printing quality become core decision factors for some users, pointing to the aesthetic and collectible value of physical books. In the social trust dimension, recommendations from authorities or trusted individuals like teachers or colleagues act as key decision drivers, especially in student groups or parent-child education scenarios, highlighting the transmission logic of trust within social networks influencing information credibility and consumption choices. Price perception exhibits duality: promotional tactics like discounts and points stimulate both repeat and first-time purchases. However, the negative association between "low price - piracy/low quality" also reveals that price strategies require quality assurance as a prerequisite to avoid degrading user experience. Moreover, unlike algorithm-dependent programmatic advertising paradigms [40], social trust mechanisms highlight the irreplaceable role of interpersonal networks in decision-making processes, particularly within information-overloaded environments.

**4.4.2.3 Dual nature of feedback behavior: emotion and supervision.** User feedback behavior possesses dual attributes: emotional venting and quality supervision. Negative reviews driven by poor experiences constitute a quality supervision mechanism for platforms and products, their content directly pointing towards optimization directions for products and services. Consumption feedback driven by nostalgic feelings, like "revisiting classics" or "switching from e-book to physical copy," implies a demand for emotional awakening of "reading memories," reflecting the cultural ritualistic feeling and emotional continuity of reading—such feedback serves both as emotional anchors for individual reading histories and provides clues for platforms to mine the long-tail value of "classic IPs." Finally, group heterogeneity was evident. Clustering results further revealed distinct subgroup characteristics: Children's reader groups require balancing content fun and educational value, suggesting optimization through "child-friendly" narrative design and home-school collaborative recommendation mechanisms. History enthusiast groups value the real-world explanatory power of historical texts, necessitating enhanced thematic depth and cross-scenario interpretation (e.g., "History + Workplace," "History + Life"). Price-sensitive groups require maintaining quality baselines during promotions to avoid the "low price - low quality" vicious cycle.

Overall, online book consumption is a closed-loop process of "Motivation Triggering → Decision Implementation → Feedback Iteration." Strategies like precise content matching, activating social trust, balancing price and quality, and

establishing negative review response mechanisms are required to synergistically achieve user need fulfillment and enhanced lifetime value.Crucially, this study's interpretation of user motivations (e.g., "cognitive empowerment," "affective bonding") constitutes neither speculative inference nor subjective conjecture. Rather, it derives from functional categorization grounded in users' recurring self-reported expressions within reviews (e.g., "provokes introspection," "developing admiration for the author"), systematically mapped against the Stimulus-Organism-Response (S-O-R) framework and Theory of Planned Behavior [41]. While event logic graphs do not directly measure psychological states, the captured causal chains (e.g., "author's unique perspectives → purchase decision") serve as observable proxy variables for behavioral intent. This methodological approach aligns with explanatory frameworks central to digital ethnography and computational social science.

## 5. Conclusions and implications

This study introduces an event logic graph (ELG) framework to reconstruct the full causal chain of online book purchasing behavior—from motivational triggers and decision execution to post-purchase feedback—by mining user reviews. Unlike conventional topic-clustering approaches that treat text as static, our method combines semantic aggregation with causal-behavioral inference, offering a dynamic micro-theoretical lens for digital cultural consumption.

The findings provide actionable pathways for digital booksellers to move beyond transactional models toward user-centric operations anchored in real usage scenarios. We identify four strategic dimensions.

Guided by Maslow's hierarchy and value co-creation theory, platforms should activate layered user motivations. For cognition-driven themes (e.g., "Fruits of Reflection"), co-developing "text-reality" interpretive content with experts transforms books into cognitive tools for problem-solving. For affective-social themes (e.g., "Author Connections"), building "Author IP Cosmoses" through brand communities fulfills users' identity and emotional needs. AI-enabled dialogues can further strengthen parasocial bonds, turning fandom into sustained engagement.

Design must address contextual pain points. For aesthetic/self-expression needs (e.g., "Scholar's Elegant Binding"), limited-edition artistic bindings with craft narratives satisfy symbolic consumption motives. In parenting contexts ("Essential Children's Books"), age-aligned "Growth Kits"—featuring AR interactions and co-reading guides—support developmental appropriateness. Companion mini-programs offer data-driven parental guidance, reducing anxiety. For loyal users ("Repeat-Purchase Recommendations"), tiered exclusives and bundled digital content deepen lock-in through reinforcement mechanisms.

Trust cascades should be leveraged via credible curation. A "Book Trust Alliance" platform—featuring educator- or expert-curated lists with referral incentives—converts social trust into engagement through reciprocity and social proof. Transparent pricing strategies (e.g., clear discount tiers, authenticity labels) reduce perceived risk. For nostalgic segments ("Nostalgic Reading Memories"), "Memory Revival Campaigns" that reward proof-of-purchase with classic reprints activate endowment effects and reinforce identity.

A full-cycle feedback response system is critical. Negative reviews should trigger automated reassurance within 2 hours, manual follow-up within 24 hours, and public improvement plans within 72 hours—turning complaints into trust-repair opportunities. To combat quality concerns ("Defective Editions"), an authenticity ecosystem—featuring traceable codes, publisher-direct zones, and tenfold counterfeit compensation—builds certainty through transparency and strong guarantees.

This work advances cultural consumption research by replacing static text-topic models with a dynamic causal-behavioral paradigm. It offers granular insights into how users navigate complex decisions in digital environments, particularly for symbolic and experience-rich products like books.Our analysis relies solely on Dangdang.com data, limiting cross-platform generalizability. Future studies should integrate offline behaviors, secondary markets, and multi-platform traces. Additionally, culturally specific metaphors (e.g., "book fragrance") may affect semantic generalizability; cross-cultural comparisons are needed. The sample also skews toward educated users, warranting inclusion of broader demographic segments.

## Supporting information

**S1 File. Reproducibility statement.** A detailed statement describing the availability of raw data, computational code, and experimental protocols to support the reproducibility of this study.
(DOCX)

**S2 File.** Fig 3. Supplementary figure showing the full structure of the event logic graph used for modeling online book user purchase behavior, including all observed events and transition paths.
(PDF)

## Author contributions

**Conceptualization:** Shiling Peng.

**Data curation:** Shiling Peng.

**Formal analysis:** Shiling Peng.

**Investigation:** Shiling Peng.

**Methodology:** Shiling Peng.

**Project administration:** Shiling Peng.

**Resources:** Shiling Peng.

**Supervision:** Bo Zhang.

**Validation:** Shiling Peng.

**Visualization:** Shiling Peng.

**Writing – original draft:** Shiling Peng.

**Writing – review & editing:** Shiling Peng.

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
