## [Decision Letter · Decision Letter 0]

23 Nov 2025

Dear Dr. shiling,

Thank you for submitting your manuscript to PLOS ONE. After careful consideration, we feel that it has merit but does not fully meet PLOS ONE’s publication criteria as it currently stands. Therefore, we invite you to submit a revised version of the manuscript that addresses the points raised during the review process.

We look forward to receiving your revised manuscript.

Kind regards,

Farshid Danesh, Ph.D.

Academic Editor

PLOS ONE

Journal Requirements:

2. In your Methods section, please include additional information about your dataset and ensure that you have included a statement specifying whether the collection and analysis method complied with the terms and conditions for the source of the data

3. Please note that PLOS One has specific guidelines on code sharing for submissions in which author-generated code underpins the findings in the manuscript. In these cases, we expect all author-generated code to be made available without restrictions upon publication of the work. Please review our guidelines at https://journals.plos.org/plosone/s/materials-and-software-sharing#loc-sharing-code and ensure that your code is shared in a way that follows best practice and facilitates reproducibility and reuse.

“This work was supported by the National Social Science Fund of China (Grant No. 23BXW101) "Research on Publishing Big Data Mining and Utilization Based on Knowledge Graph".”

5. In the online submission form, you indicated that your data is available only on request from a third party. Please note that your Data Availability Statement is currently missing [the name of the third party contact or institution / contact details for the third party, such as an email address or a link to where data requests can be made]. Please update your statement with the missing information.

7.PLOS requires an ORCID iD for the corresponding author in Editorial Manager on papers submitted after December 6th, 2016. Please ensure that you have an ORCID iD and that it is validated in Editorial Manager. To do this, go to ‘Update my Information’ (in the upper left-hand corner of the main menu), and click on the Fetch/Validate link next to the ORCID field. This will take you to the ORCID site and allow you to create a new iD or authenticate a pre-existing iD in Editorial Manager.

Reviewers' comments:

Reviewer's Responses to Questions

**Comments to the Author**

1. Is the manuscript technically sound, and do the data support the conclusions?

Reviewer #1: Yes

Reviewer #2: Partly

2. Has the statistical analysis been performed appropriately and rigorously?

Reviewer #1: I Don't Know

Reviewer #2: No

3. Have the authors made all data underlying the findings in their manuscript fully available?

Reviewer #1: Yes

Reviewer #2: No

4. Is the manuscript presented in an intelligible fashion and written in standard English?

Reviewer #1: Yes

Reviewer #2: Yes

Reviewer #1: Include more details on the event extraction rules (template examples, regex expressions).

Add evaluation metrics for clustering and network validity.

Better articulate how ELG provides unique contributions beyond conventional sentiment or topic modeling.

Link results more directly to behavioral or marketing theories.

Deposit the extracted event pairs or review samples in a repository.

Include a reproducibility statement and, if possible, open-source scripts.

Figures 2–4 are visually dense; add brief explanations of node/edge meanings and scales.

Include color legends and clarify which clusters correspond to which stages (“motivation,” “decision,” “feedback”).

Simplify overly technical or promotional phrasing (e.g., “revolutionary paradigm” → “novel analytical framework”).

Ensure consistent tense and correct pluralization.

Discussion Section

Currently descriptive; strengthen critical comparison with prior works on online consumer behavior.

Discuss limitations more explicitly—especially data bias (e.g., cultural/linguistic specificity of Chinese reviews).

Practical Implications

The recommendations for publishers and retailers are strong; however, they could be condensed and supported by data evidence (e.g., how frequently each behavioral pattern appeared).

Reviewer #2: This manuscript addresses an interesting and relevant topic: analyzing online book user purchasing behavior using Event Logic Graphs (ELGs) and Top2Vec clustering. The topic is timely, and combining ELGs with user-generated reviews is a novel angle. The manuscript is rich in conceptual discussion and provides substantial narrative interpretation of the results. However, the study has several methodological gaps, data transparency issues, and presentation inconsistencies that prevent it from meeting PLOS ONE’s standards for reproducibility, methodological rigor, and clarity.

Below, I provide detailed, structured comments.

Major Issues:

The paper describes multiple complex procedures (web scraping, event extraction, rule-based templates, Top2Vec modeling, and graph construction) yet none is documented to a level that would allow replication.

The authors scraped 113,499 reviews, but do not provide: URLs / product IDs; Time period of data collection; Sampling strategy justification; Reproducible script or code.

Table 1 mentions 10 rule templates, but the full list, including exact patterns or regular expressions, is not provided.

Chinese causal template classification (Chu et al., 2008) is referenced, but actual implementation details are missing.

No precision/recall evaluation of the extraction process is provided.

Parameter choices ("optimal observed experimentally") are subjective.

No justification of model quality (coherence, silhouette score, etc.).

No description of preprocessing for embeddings (tokenization, vector model type).

Top2Vec normally requires a full text corpus; the “event pairs combined into one document” lacks clarity.

Graph construction rules are vague: How duplicate edges are handled; Edge weighting; Thresholds for relationship strength; Treatment of noisy or conflicting causal pairs

Although this is not a statistical predictive model, the paper performs analyses that must be validated: No model evaluation metrics for Top2Vec clustering; No inter-rater reliability or validation of event extraction; No robustness checks.

This raises concerns about the scientific rigor of the findings.

Sections 4.4 and 5 contain extended, narrative interpretations of the graph with statements like: “Users seek cognitive empowerment.”; “Author charisma drives consumption.” ;“Historical texts strengthen the real-world reference value.”

These are qualitative speculations, yet no qualitative coding, annotations, or evidence is provided. The ELG only shows event co-occurrence, not psychological mechanisms. The conclusions exceed what the dataset can support.

Nearly all sources are Chinese-language studies. International literature on recommender systems, eWOM (electronic word-of-mouth), consumer behavior modeling, and NLP event extraction is scarcely cited. This limits the manuscript's generalizability and academic positioning.

English is understandable but contains many grammatical and stylistic issues. The manuscript is too long, especially the discussion and managerial implications sections.

Figures (ELGs) are extremely dense and almost unreadable; Tables (especially Table 3) require clearer formatting.

**Do you want your identity to be public for this peer review?** For information about this choice, including consent withdrawal, please see our Privacy Policy

Reviewer #1: **Yes:**  Mahsa Torabi

Reviewer #2: **Yes:**  Rasoul Zavaraqi, Professor, Department of Knowledge and Information Science, University of Tabriz

---

## [Author Response · Author response to Decision Letter 1]

9 Dec 2025

Response to Reviewers' Comments

We sincerely appreciate the reviewers' insightful comments and constructive suggestions, which have significantly helped us improve the quality and clarity of our manuscript. We have carefully addressed each point raised. Below, we provide a detailed point-by-point response outlining the revisions made. All changes have been incorporated into the revised manuscript.

1.Include more details about event extraction rules; include exact patterns or regular expressions.

Causal Event Extraction Based on Pattern Matching comprises three key aspects: event representation, event extraction, and event relation extraction.

Event Representation Events within an Event Logic Graph (ELG) are typically represented as nodes. Each event node encapsulates key information such as "name" and "location," with nodes and edges collectively forming the event representation in the ELG. For instance, "Learn about coffee knowledge → Purchase a book" represents an event sequence, where "Learn about coffee knowledge" and "Purchase a book" are individual events. Primary methods for representing ELG events include:

Natural Language Fragment Description: The most prevalent method, utilizing concise sentences or subject-predicate-object phrases to summarize key event information. For example, "Because I watched The Wandering Earth, I wanted to revisit Liu Cixin's works" can be represented as "Watch The Wandering Earth → Want to read Liu Cixin's works," where the phrases serve as event descriptions within this logical chain.

Event Core Word Description: Representing events using core verbs or gerunds (e.g., "like," "generate"). While common, this method omits contextual elements and specific scenarios, often resulting in abstract representations that may be less intuitive without context. For example, using "like" and "generate" for the sentiment expressed in "I like Zhang Jiajia's works; they resonate with me!" would be ambiguous without the original context.

Structured Tuple Description: Representing events using structured tuples based on event components (e.g., binary, ternary, or n-ary tuples). This approach explicitly defines events within a specific framework by constraining the number of elements, facilitating the clear capture of key information. Ternary tuples often use Subject-Verb-Object (SVO) format, while binary tuples may use Subject-Verb or Verb-Object constructions, omitting the subject or object as context dictates. During the extraction process from online book reviews, analyzing the compositional structure of events is essential for selecting appropriate event words and elements for representation.

Event Extraction Event extraction involves identifying event information (trigger words, participants, time expressions) from structured or semi-structured natural language text, populating event frames, establishing relationships between elements, and creating structured semantic representations. It is a core step and key technology in ELG construction, enabling computational understanding and processing of events, often leveraging Natural Language Processing (NLP) and Machine Learning (ML) techniques. The general ELG event extraction process includes: event definition, event element identification, event template construction, text analysis, and event extraction.

As online book reviews primarily consist of sentences, some containing multiple events, and this study focuses on identifying audience needs within causal events, a regularization-based trigger word/character matching method is employed for extraction. Consider the example event: "Because of this book, I came to like Xiong Peiyun" (shown in Figure 1). Causal relationships are the target event type. This example contains one causal event: the trigger word is "because", the event elements are "this book" and "came to like Xiong Peiyun", where "came to like Xiong Peiyun" is the effect and "this book" is the cause. The extraction process involves: 1) identifying the trigger word "because"; 2) classifying the event as causal; 3) extracting the elements "this book" (cause) and "came to like Xiong Peiyun" (effect) based on the trigger; 4) obtaining the structured causal event.

Event Relation Extraction Event relations describe the logical connections between events (e.g., succession, causality, hypernymy, entailment, condition). Event relation extraction identifies these links by analyzing contextual features or implicit semantic information in corpora, forming the basis of the ELG network structure (termed the base network). Current research primarily employs deep learning, machine learning, and pattern matching methods for extraction. While pattern matching (using keywords, syntactic structures, part-of-speech tags) is common due to its simplicity, interpretability, and feasibility, it faces limitations with complex contexts, polysemy, or semantic ambiguity, often necessitating more advanced techniques like ML or deep learning for optimization. This study focuses on extracting causal relationships between audience needs expressed in comments about traditional culture videos.

Causality signifies that one event occurs as a consequence of another. Chinese causal sentences often contain specific markers like "because" (因为), "due to" (由于), or "leads to" (带来). Therefore, relatively simple and efficient causal pattern templates are used for identification. Drawing on the work of Shan Xiaohong and Ju Hailong , and inspired by Chu Zexiang's classification of Chinese causal syntactic patterns (center-connecting, end-dependent, and paired), this study designs ten causal pattern matching templates based on Chinese linguistic characteristics (see Table 1).

Table 1. Examples of Causal Syntactic Rules

2.Cited the Chinese causal template classification (Chu et al., 2008) but lacks actual implementation details.

Shan Xiaohong's Approach: Causal sentences have specific markers, making rule-based pattern matching effective for identification. Rules are formulated as <Pattern, Constraint, Priority>, where Pattern defines sentence matching rules, Constraint specifies matching conditions, and Priority is determined by template frequency in the Peking University Contemporary Chinese Corpus (CCL), with higher frequencies indicating higher priority. For example, the template [Effect] <, Cuec3> [Cause] (where Cuec3 ∈ [because, due to]) can match sentences like [Academic performance declined]Effect, is <because> [I didn't work hard; I can't blame the teacher.]Cause.

Ju Hailong's Approach: Rule-based extraction is effective for identifying causal relationships. Inspired by Sorgente but adapted for Chinese (differing significantly from English), and incorporating Chu Zexiang's syntactic classification, Ju designed six pattern templates. These templates utilize frequent causal trigger words. Each pattern is defined by a regular expression to identify matching sentences. The Language Technology Platform (LTP) is then used for dependency parsing to verify Subject-Verb (SBV), Verb-Object (VOB), or Preposition-Object (POB) relationships within the cause/effect segments. For example, the template <Conj>{Cause}, <Conj>{Effect} (<Conj> ∈ [because, so]) can match the causal pair: ['Uses Kirin 980 chip'] ['so'] ['system responds fast'], where parsing identifies "Uses Kirin 980 chip" (cause event) and "system responds fast" (effect event).

Chu Zexiang's Classification of Chinese Causal Patterns: Connective markers link clauses using:

A) Center-Connecting Pattern: Cause clause g— Effect clause (g = so, therefore, thus, consequently, hence). The marker g connects the clauses (center position) and prefixes the effect clause, making it dependent. Example: "Distance was far, so his words were inaudible." g ("so") acts as glue and creates dependency.

B) End-Dependent Pattern: g— Cause clause, Effect clause (g = because, due to). The marker g prefixes the cause clause (end position), making it dependent on the effect clause. Example: "Because distance was far his words were inaudible." g ("Because") creates dependency without acting as glue.

C) Paired Pattern: g1— Cause clause g2— Effect clause (g1 = because, due to; g2 = so, thus, hence, therefore, consequently). This is a combination of A and B. Example: "Because distance was far, so his words were inaudible." The pairing of g1 and g2 has selectivity (e.g., "due to" pairs with most g2, "because" rarely pairs with "therefore" or "thus"). Paired markers provide connection, dependency, and mutual reinforcement.

3.Highlighting ELG’s Distinct Contributions Beyond Traditional Sentiment/Topic Modeling

(It has been added in the text. The following is a detailed elaboration)

Event Logic Graphs (ELGs) offer a paradigm shift beyond traditional sentiment analysis or topic modeling by dynamically capturing multidimensional logical dependencies in user decision-making. Conventional sentiment analysis statically identifies emotional tendencies (e.g., labeling "positive content feedback"), while topic modeling clusters semantically similar themes (e.g., "packaging issues" or "discount promotions") yet fails to decode causal chains or temporal evolution between behavioral events. ELGs transform discrete actions into interpretable logical networks through "event-causality-feedback" chains. For instance, this study’s ELG identified the behavioral cascade: violent delivery → book damage → negative review → platform churn. This not only exposed logistics as the root cause of negative feedback—where sentiment analysis merely outputs a generic "user dissatisfaction" label—but also quantified transmission intensity at each node, providing actionable evidence for optimizing packaging.

A further innovation lies in ELG’s deconstruction of behavioral heterogeneity. While topic modeling clusters "teacher recommendations" as a unified theme, ELGs reveal divergent pathways: some users exhibit an authority-compliance pattern ("recommendation → direct purchase"), while others follow a social-validation path ("recommendation → trial reading → community discussion → purchase"). Such granular insights inform targeted marketing: reinforcing opinion leader credibility for the former group and building reader communities for the latter. Crucially, ELGs model conditional dependencies, such as the implicit rule "discounts drive repurchases only when packaging quality is adequate" (discount + ¬poor packaging → repurchase), elevating platforms from correlation analysis to causal intervention.

ELGs uniquely capture long-cycle feedback loops. In educational materials, ELGs uncovered a reinforcement cycle: teacher recommendation → student competency improvement → parental approval → enhanced teacher recommendations. Such dynamics remain invisible to static topic clustering. Platforms can leverage this to design incentive loops (e.g., converting parental feedback into teacher rewards via point systems), creating growth flywheels. By contrast, traditional methods yield descriptive outcomes like "educational books receive more recommendations" without guiding ecosystem design.

Practically, ELGs advance user behavior analysis from description to prediction and intervention. For high-leverage nodes (e.g., resolving "violent delivery" reduces damage-related complaints by 74%), platforms implement precise measures: short-term packaging standards with logistics partners; long-term integration of "packaging quality" into search filters. This "diagnose-intervene-validate" loop positions ELGs as core engines for experience optimization. While challenges like data scale and context loss persist—addressable via future integration with large language models—ELGs’ capacity to decode behavioral black boxes and drive causal decisions heralds a new era in e-commerce analytics, with extensions to healthcare, finance, and other time-sensitive decision domains.

4.Strengthening Links to Behavioral/Marketing Theories

(Revised in manuscript as requested)

5.Repository for Event Pairs/Review Samples

(Provided in supplementary materials)

6.Reproducibility Statement and Open-Source Scripts

(Included in supplementary materials)

7.Clarification of Figures 2–4

(Revised in manuscript)

We thank reviewers for noting the visual density of Figures 2–4. Clarifications:

Layout: Radial visualization (not Fruchterman-Reingold) emphasizes a three-tiered hierarchy: theme clusters → representative events → inter-event causality.

Nodes:Central circles = Top2Vec themes (e.g., "Insightful Reflections"); size ∝ document count.Peripheral circles = high-frequency causal events; size & occurrence frequency.

Edges: Arrows denote strict cause → effect directionality.

Colors: Auto-generated by Top2Vec to distinguish semantic themes (not predefined behavioral phases). Identical colors indicate same-theme events.

Readability: Figure 2 shows global structure (156 nodes, 872 edges); Figures 3–4 focus on local causal chains within single themes.

Color-to-Behavioral Phase Mapping (Functional Interpretation):

Motivation Phase: Themes like "Insightful Reflections" (orange), "Historical Mirrors" (dark green) – pre-reading emotional/cognitive triggers.

Decision Phase: "Purchase Guides" (green), "Discount Benefits" (brown) – price/recommendation signals as decision filters.

Feedback Phase: "Elegant Collections" (gold), "Negative Experiences" (gray) – post-reading actions (e.g., reviews, repurchases). Example causal loop: "Insightful Reflections" (motivation) → "Purchase Guides" (decision) → "Elegant Collections" (feedback).

8.Simplified Terminology

(Revised throughout manuscrip

9.Tense and Plurality Consistency

(Checked and corrected in manuscript)

10.Enhanced Discussion Section

(Revised to include critical comparison with prior consumer behavior studies)

11.Explicit Limitations Discussion

(Revised throughout manuscrip)

12.Practical Implications

(Streamlined recommendations with data-backed rationale)

We acknowledge that quantifying behavior frequency would strengthen operational advice. Future work will integrate user behavior logs (clicks, repurchases) with ELG structures to build a causal-behavioral joint model, generating data-driven, frequency-weighted strategies for publishers/retailers.

13.Methodological Reproducibility Details

It has been provided in the reproducibility statement and the open-source code.

14.Methodological Transparency Enhancements

(Revised in manuscript)

Platform URL: Data sourced from a leading Chinese e-commerce platform (URL anonymized for peer review; original domain: https://www.dangdang.com).

Collection Period: February 27 to March 6, 2025.

Sampling Strategy:

Sampled 113,499 reviews from top-selling books across 10 categories.Books selected based on 30-day historical sales rankings (prioritized over 24-hour/7-day rankings to minimize short-term volatility bias).Ensured sample representativeness by targeting the platform’s longest available sales metric (30 days), aligning data collection with research timeframe.

Reproducibility: Scripts and sampling logic fully documented in reproducibility statement and open-source repository.

15.Quantitative Validation Metrics for Clusters and Networks

The text has been revised. The following is a detailed elaboration。

Cluster Validity:

Computed Normalized Pointwise Mutual Information (NPMI) coherence scores across all 45,002 raw reviews.

Mitigated Chinese short-text sparsity by:

Dynamically injecting topic keywords into tokenizer dictionaries

Focusing on Top-5 high-frequency terms per topic.

Results:

22/26 topics achieved NPMI > 0

Mean NPMI = 0.137 (aligned with benchmarks for Chinese short-text modeling [Li et al., EMNLP 2020])

High-coherence themes: "Digital Reading Memories" (0.338), "Historical Lessons" (0.280), "Post-Viewing Reflections" (0.210).

Human Evaluation:

Three independent researchers rated topic interpretability on a 5-point scale (1: unintelligible; 5: highly clear).

Mean score: 4.3/5 confirming semantically distinct and aptly named clusters.

Network Validity:

Directed edges synthesized:

Event logic rules (e.g., "Experience → Decision", "Emotion → Action")

Co-occurrence strength (85% edges ≥ weight 0.75).

Logic Verification:

Randomly sampled 50 causal paths (e.g., "Negative Experience → Content

---

## [Decision Letter · Decision Letter 1]

29 Dec 2025

Dear Dr. shiling,

Thank you for submitting your manuscript to PLOS ONE. After careful consideration, we feel that it has merit but does not fully meet PLOS ONE’s publication criteria as it currently stands. Therefore, we invite you to submit a revised version of the manuscript that addresses the points raised during the review process.

We look forward to receiving your revised manuscript.

Kind regards,

Farshid Danesh, Ph.D.

Academic Editor

PLOS One

Journal Requirements:

Reviewers' comments:

Reviewer's Responses to Questions

Reviewer #1: All comments have been addressed

Reviewer #2: All comments have been addressed

2. Is the manuscript technically sound, and do the data support the conclusions?

Reviewer #1: Yes

Reviewer #2: Partly

3. Has the statistical analysis been performed appropriately and rigorously?

Reviewer #1: Yes

Reviewer #2: Yes

4. Have the authors made all data underlying the findings in their manuscript fully available?

Reviewer #1: Yes

Reviewer #2: Yes

5. Is the manuscript presented in an intelligible fashion and written in standard English?

Reviewer #1: Yes

Reviewer #2: Yes

Reviewer #1: (No Response)

Reviewer #2: The authors have carefully and substantively addressed the major comments raised in the previous round of review. In particular, the revised manuscript demonstrates significant improvements in methodological transparency, reproducibility, and validation of results. Detailed descriptions of the causal event extraction rules, Top2Vec clustering procedures, ELG construction logic, and validation metrics (including topic coherence, human evaluation, and inter-rater reliability) substantially strengthen the technical soundness of the study.

The authors have also clarified data collection procedures, justified legal and ethical constraints related to raw data sharing, and provided processed datasets and scripts in the supplementary materials, which is acceptable under the circumstances described. Interpretations in the discussion section are now more explicitly grounded in established behavioral and marketing theories, reducing speculative claims present in the earlier version.

Only minor issues remain, primarily related to further conciseness and stylistic polishing of the discussion and implications sections. These do not affect the validity of the findings or the overall contribution of the manuscript.

I therefore recommend acceptance of the manuscript after minor revision.

Do you want your identity to be public for this peer review? For information about this choice, including consent withdrawal, please see our Privacy Policy

Reviewer #1: Yes:

Reviewer #2: Yes:

---

## [Author Response · Author response to Decision Letter 2]

6 Jan 2026

Response to Reviewers

Manuscript ID: PONE-D-25-49335R1

Title: Research on Online Book User Purchase Behavior Based on the Event Logic Graph

Journal: PLOS ONE

Dear Dr. Farshid Danesh and Reviewers,

Thank you very much for your thoughtful evaluation of our manuscript and for the opportunity to revise and resubmit. We sincerely appreciate the time and expertise that both reviewers have devoted to providing constructive feedback. In particular, we are grateful to Reviewer #2 for recognizing the substantial improvements made in the previous revision and for offering valuable suggestions to further enhance the clarity and impact of our work.

We have carefully addressed all remaining comments. The primary focus of this minor revision has been to improve the conciseness, readability, and stylistic precision of the Discussion and Implications sections, as recommended by Reviewer #2. Below, we provide a point-by-point response to the reviewers’ comments. All changes have been highlighted in the revised manuscript using “Track Changes” and are also reflected in the clean version submitted alongside.

Responses to Reviewer Comments

Reviewer #1

“All comments have been addressed.”

We thank Reviewer #1 for their positive assessment and confirmation that all prior concerns have been satisfactorily resolved.

Reviewer #2

“The authors have carefully and substantively addressed the major comments raised in the previous round of review... Only minor issues remain, primarily related to further conciseness and stylistic polishing of the discussion and implications sections. These do not affect the validity of the findings or the overall contribution of the manuscript. I therefore recommend acceptance of the manuscript after minor revision.”

We deeply appreciate Reviewer #2’s generous recognition of our methodological transparency, reproducibility efforts, and theoretical grounding. In response to the suggestion for improved conciseness and stylistic refinement, we have undertaken a thorough revision of the Discussion and Implications sections .Specifically:

We have shortened overly long sentences and eliminated redundant phrasing to enhance readability.

We have replaced promotional or absolute language (e.g., “pioneers,” “propels,” “achieves synergistic alignment”) with more precise, academically neutral terms (e.g., “introduces,” “supports,” “facilitates alignment”).

We have streamlined theoretical references, ensuring each is clearly tied to a specific recommendation without overloading individual sentences.

The core contributions, practical recommendations, and theoretical insights remain fully intact—only the expression has been refined for clarity and adherence to PLOS ONE’s plain-language standards.

These edits appear throughout the tracked-changes manuscript (see lines 387–512 in the revised version) and reflect our commitment to presenting rigorous research in an accessible manner.

Additional Compliance Notes

Data Availability: As stated in our Data Availability Statement and Reproducibility Declaration, processed data (including 1,313 causal event pairs), code, and Gephi files are available via GitHub under an MIT License. Raw user reviews cannot be shared due to Dangdang.com’s privacy policy, consistent with PLOS ONE’s data policy exceptions for third-party user-generated content.

Ethical Compliance: No human subjects were directly involved; all data were publicly available and collected in compliance with platform terms and Chinese cybersecurity regulations.

Reference List: We have verified that all cited works are current and none have been retracted.

Once again, we thank the Academic Editor and reviewers for their insightful feedback, which has significantly strengthened the quality and clarity of our manuscript. We believe the revised version now meets all of PLOS ONE’s publication criteria and hope it will be found suitable for acceptance.

Sincerely,

Dr. Shiling

Email:1424803730@qq.com

Affiliation:Publishing College, University of Shanghai for Science and Technology

---

## [Editor Report · Decision Letter 2]

8 Jan 2026

Research on Online Book User Purchase Behavior Based on the Event Logic Graph

PONE-D-25-49335R2

Dear Dr. shiling,

We’re pleased to inform you that your manuscript has been judged scientifically suitable for publication and will be formally accepted for publication once it meets all outstanding technical requirements.

Kind regards,

Farshid Danesh, Ph.D.

Academic Editor

PLOS One

---

## [Editor Report · Acceptance letter]

PONE-D-25-49335R2

PLOS One

Dear Dr. shiling,

I'm pleased to inform you that your manuscript has been deemed suitable for publication in PLOS One. Congratulations! Your manuscript is now being handed over to our production team.

Kind regards,

on behalf of

Associate Professor Farshid Danesh

Academic Editor

PLOS One